# Carbon Footprint Analysis of Sewage Sludge Thermochemical Conversion Technologies

**Liping Li** [1,2], **Guiyue Du** [1,2], **Beibei Yan** [1,2], **Yuan Wang** [1,2], **Yingxin Zhao** [1,2], **Jianming Su** [3], **Hongyi Li** [3], **Yanfeng Du** [3], **Yunan Sun** [4,*], **Guanyi Chen** [1,2,4], **Wanqing Li** [4] and **Thomas Helmer Pedersen** [5]

1　School of Environmental Science and Engineering, Tianjin University, Tianjin 300072, China
2　Tianjin Key Lab of Biomass/Wastes Utilization, Tianjin University, Tianjin 300072, China
3　Tianjin Huabo Water Affairs Co., Ltd., Tianjin 300301, China
4　School of Mechanical Engineering, Tianjin University of Commerce, Tianjin 300134, China
5　Department of Energy Technology, Aalborg University, 9220 Aalborg, Denmark
*　Correspondence: sunyunan@tjcu.edu.cn

**Abstract:** Thermochemical conversion technology for sewage sludge (SS) management has obvious advantages compared to traditional technologies, such as considerable volume reduction, effective pathogen elimination, and potential fuel production. However, few researchers conducted comparative research on the greenhouse gas (GHG) emission performances of these technologies. This paper evaluates the lifecycle carbon footprints of three SS thermochemical conversion technologies, including hydrothermal liquefaction (HTL) (Case 1), pyrolysis (Case 2), and incineration (Case 3) with software OpenLCA and Ecoinvent database. The results show that Case 1 has the smallest carbon footprint (172.50 kg $CO_{2eq}$/t SS), which indicates the HTL process has the best GHG emission reduction potential compared to other SS disposal routes. The biggest contributor to the carbon footprint of SS thermochemical conversion technologies is indirect emissions related to energy consumption. So the energy consumption ratio (ECR) of the three cases is calculated to assess the energy consumption performances. From the perspective of energy conversion, Case 1 shows the best performance with an ECR of 0.34. In addition, element balance analysis is carried out to deeply evaluate the carbon reduction performance of the three cases. This study fills the knowledge gap regarding the carbon footprints for SS thermochemical conversion technologies and provides a reference for future technology selection and policymaking against climate change in the SS management sector.

**Keywords:** sewage sludge; carbon footprint; energy consumption; thermochemical conversion

## 1. Introduction

To deal with global climate change and limit temperature increase, as agreed upon in the Paris Agreement [1,2], China is taking action to fulfill its goal of carbon neutrality by 2060. With rapid urbanization and population growth, the continuous increase in sewage sludge (SS) production from wastewater treatment plants (WWTPs) has become a big challenge to meeting greenhouse gas (GHG) reduction goals. SS is considered the vital byproduct of WWTPs, mainly containing primary sludge and waste-activated sludge when the activated sludge process is used. It was estimated that each person produced 0.1–30.8 kg SS per year on average [3–5]. In China, the SS yield related to WWTPs had increased to up to $10^5$ t/day [6]. Considering the consistent economic growth and increase in population and urbanization, a continued rising trend for SS production is expected in the future.

In general, SS is a mixture of microorganisms and zoogloea granules. The moisture content of SS usually exceeds 90 wt% after treated by clarifier and thickener [7]. Due to the characteristic of SS, it takes a massive amount of energy to remove the water and make it favorable for further processing. For a WWTP, the operation associated with SS treatment and disposal may make up more than 50% of its operating expenses and contribute to 40%

of its GHG emissions [8]. To meet the target of carbon neutrality for the globe and for China, understanding the GHG emission performances of sludge management technologies is crucial to decision making within the wastewater treatment sector.

Anaerobic digestion, aerobic composting, and landfill were methods that were once applied widely for SS treatment and disposal. However, harmful pollutants such as pathogenic agents, heavy metals, and pharmaceuticals in SS can not be destroyed and removed completely with the above disposal routes, which causes problems for human health and the environment. Complete elimination of pathogens is especially important in the context of the global COVID-19 epidemic. These methods also take a long time and a large area of land space to decompose the organic matter in SS [9–18]. Another drawback of these methods is the waste of carbon. It is widely known that during microorganism degradation in landfill and agricultural application processes, the carbon in SS is converted to carbonaceous gas that dissipates in the atmosphere, and causes odor and GHG emission. Even in the anaerobic digestion process, only 20–30 wt% of the carbon in SS is recycled as $CH_4$ to substitute energy consumption; the remaining 70–75 wt% is converted to $CO_2$ and is emitted into the environment [7].

The thermochemical conversion technology for SS management has obvious advantages compared to the above-mentioned traditional technologies. Through the conversion of carbon in SS to fuel gas, bio-oil, and biochar, this method achieves not only considerable volume reduction but also effective pathogen elimination, potential fuel production, and unstructured GHG emission mitigation [19,20]. With the increasing number of wastewater treatment infrastructures around the world, thermochemical conversion of SS has been deemed one of the most promising technologies to handle a growing volume of SS [19,21–24].

A growing number of investigations on thermochemical conversion of SS have been reported. SS incineration technology is quite mature and widely used around the world. Chen et al. [18] carried out the environmental, energy, and economic assessment of the co-incineration of municipal solid waste and SS in China. In addition, hydrothermal liquefaction (HTL), gasification, and pyrolysis of SS are still in the lab or pilot scale, and the configuration and running parameters need to be further optimized [7,25,26]. Meanwhile, the energy efficiency, economic efficiency, and environmental impacts of SS thermochemical conversion technologies received more attention. Life cycle assessment(LCA) has been applied extensively as an useful tool for economic and environmental benefits evaluation [27–29]. However, most of the research focused on energy recovery efficiency or costs, while little comparative research was conducted on the GHG emission performances of different SS thermochemical conversion routes.

To fill this knowledge gap, three SS thermochemical conversion technologies, including HTL, pyrolysis, and incineration were compared based on their carbon footprints, energy consumption, and element balances using the methodology of life cycle assessment(LCA), energy consumption ratio (ECR),and material flow analysis (MFA) in this study. The results provide a scientific basis for technology selection and policy planning of SS management to reduce climate change.

## 2. Materials and Methods

### 2.1. Case Study Descriptions

Thermochemical conversion technologies can be sorted into two groups based on their requirements for feedstock moisture content. Usually, a feedstock with a moisture content below 10wt% can be treated through incineration and pyrolysis processes. However, the pre-drying step for reducing the moisture content in wet feedstock requires a large quantity of energy, which greatly affects the GHG emissions related to energy consumption. In contrast, the hydrothermal method can deal with wet feedstock without the energy-consuming step of drying and produce similar products [30]. Therefore, three cases were selected: Case 1 (HTL), which represents non-pre-drying technologies, Case 2

(pyrolysis), and Case 3 (incineration). Cases 2 and 3 represent pre-drying technologies in different degrees.

In WWTPs of China, the mixture of primary sludge and waste-activated sludge is commonly thickened and dewatered to a moisture content of 80wt% or so, and then the dewatered sludge is transported to a specialized plant for subsequent processing and final disposal. Hence, dewatered sludge with a moisture of 80 wt% was used as the uniform feedstock in all three cases.

Case 1. Sludge slurry was processed via HTL at 340 °C for 20 min. Then the reactor was cooled to room temperature and the products of fuel gas, water, bio-oil, and solid phase were collected and separated for weighing. The gas product was discharged into the atmosphere without extra treatment. The bio-oil was recovered as an alternative energy source, and the solid was landfilled. The aqueous phase was sent back to the reactor for cyclic utilization.

Case 2. SS was pre-dried to a moisture content of 7 wt%, while the rest of the wastewater was collected and sent back to the WWTP for treatment. Then, the dried sludge was sent to a fluidized bubbling bed reactor for pyrolysis. The reactor was heated to 500 °C with nitrogen as the fluidization gas. After the reactor cooled to room temperature, the bio-oil and biochar yields were determined by weighing, and the gas yield was calculated from the difference. All the products were collected for substituting fossil energy.

Case 3. As is known to all, SS with a moisture content of 40 wt% shows almost the same combustion characteristics as those with a moisture content of 10 wt%, but requires less energy, and economic and environmental cost [18,31,32]. In case 3, the moisture content in SS was removed through a heating process, while the rest of the wastewater was collected in a sewage tank and sent back to the WWTP. Semi-dried sludge with a moisture content of 40 wt% was sent to the circulating fluidized bed incinerator for power generation. Diesel was added in the ignition stage to make the SS burn. To help the SS burn steadily in the incinerator, coal was added as auxiliary fuel to offset the low heating value of SS. The incinerating process converted the moisture and combustible composition in SS to flue gas, which followed the system and entered the flue gas purification system (with $CaO$) with a bag filter to remove the acid gases and dust, etc. In the end, the non-combustible compositions in SS were delivered to landfill.

### 2.2. Methodology for Carbon Footprint Accounting

The LCA provides a scientific and useful tool for quantitatively evaluating resource consumption and environmental emission of different products and technologies during a period or for their whole lifetime [33]. The carbon footprint accounting of the SS thermochemical conversion technologies followed the theory and procedure of the LCA. The definition of carbon footprint in this study was the summation of direct GHG emissions in the sludge treatment unit and the indirect GHG emissions caused by energy and chemical consumptions within the system boundary. The GHGs calculated in the study included $CO_2$, $CH_4$, and $N_2O$. For comparison, the latter two kinds of gases were converted by global warming potentials (GWPs) of 28 and 265 respectively into carbon dioxide equivalents ($CO_{2eq}$).

This part of the study aimed to estimate and compare the carbon footprint of the three chosen SS thermochemical conversion cases. A further goal was to identify the most important contributor to GHG emissions.

OpenLCA, created by GreenDelta, is an open-source LCA software used for creating the model for each case in the research. Ecoinvent, an European reference Life Cycle Database of the Joint Research Center, was used as the background database and is widely applied in previous studies. IPCC 2013 (GWP100) was selected as the impact assessment method in OpenLCA. The next sections detail the setting of the functional unit and system boundary as well as the interpretation of inventorydata and results.

### 2.2.1. Functional Unit

Different from the LCA of a product, the functional unit in a waste disposal LCA is usually defined as the input for the system. In this study, the functional unit was the disposal of 1 t dewatered SS with a moisture content of 80 wt%. In the life cycle inventory, all the consumption, production, and emission data were adapted per functional unit (t SS).

### 2.2.2. System Boundary

Figure 1 indicates the system boundaries of the three cases. The system boundary started from SS drying (if necessary) and ended at the output of useful products from the thermochemical conversion process. Previous research results showed that the environmental impacts of the construction and demolition phases were much less than those of the operation period [34–36], and were not accounted for in this study. The GHG emissions related to the energy consumption of SS dewatering from moisture of 95 wt% to 80 wt% and transportation from the WWTP to the sludge disposal plant were attributed to the WWTP, which was not included in this assessment. The GHG emission in the wastewater plant was not considered since it was assumed to be the same for all the cases.

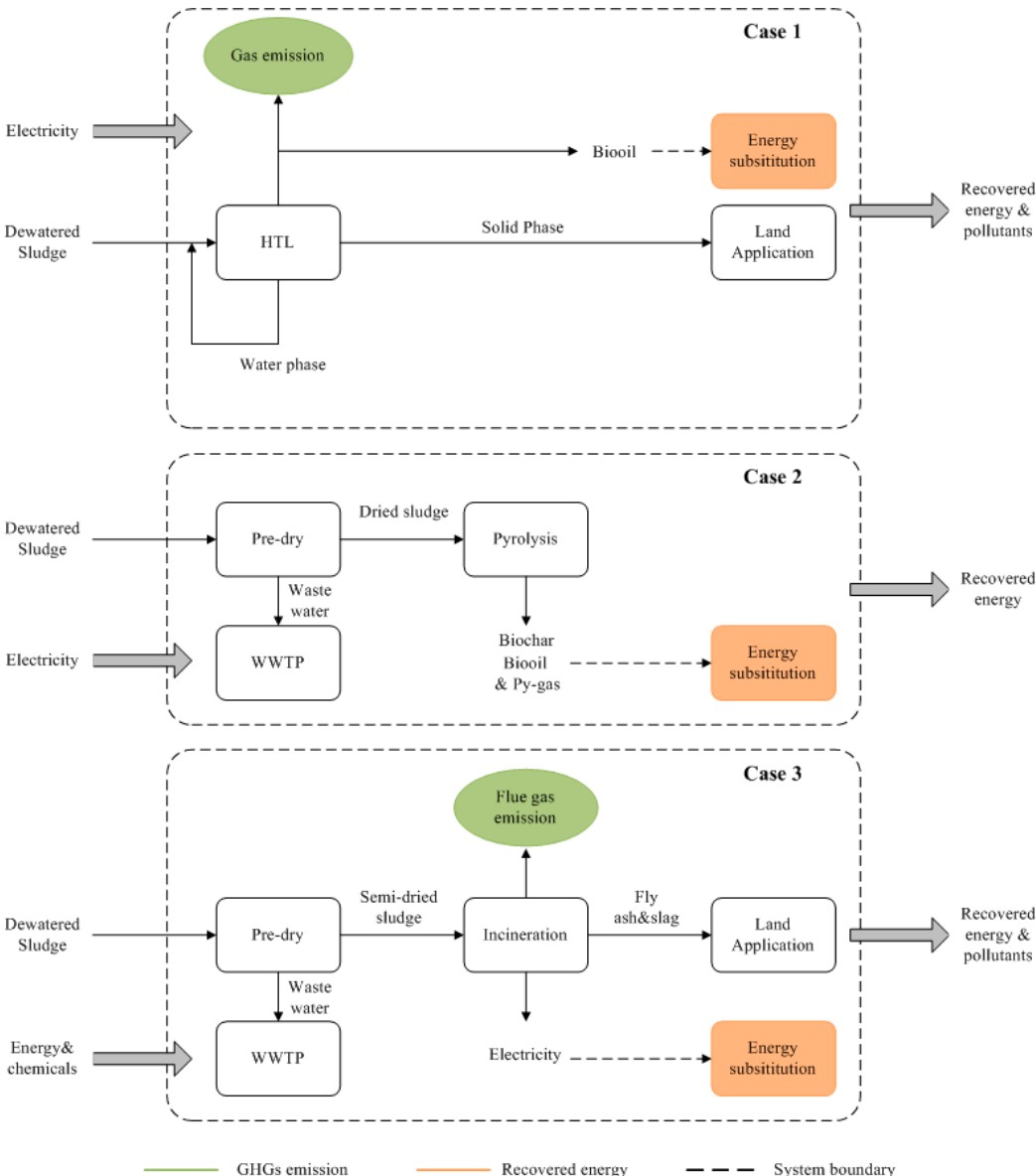

**Figure 1.** System boundaries of the three cases.

Within the defined system boundary of the three cases, SS, energy (electricity, coal, and diesel) and chemicals (e.g., CaO, $H_2O$, etc.) were considered as input; energy products (fuel gas, bio-oil, biochar, and electricity), GHGs and other contaminants were output. Direct GHG emissions from sludge treatment (e.g., $CH_4$ emission from HTL), indirect emissions from the production and transportation of chemicals, electricity, and auxiliary fuel consumption during operations were included in the carbon footprint calculation and evaluation. For convenient comparison, the substitution effects of the energy products are evaluated by converting them to an equivalent amount of heat or electricity. At the end of the process for Cases 1 and 3, the solid residuals, such as biochar, were delivered to a sanitary landfill, but the energy consumption and related GHG emissions were not taken into account in this paper [37,38].

It is worth noting that due to its biogenic origin, GHG inventory by Intergovernmental Panel on Climate Change (IPCC) does not contain the direct $CO_2$ emissions from the SS treatment and disposal process. However, many researchers found that carbon derived from fossils can take up to approximately 20 wt% of total carbon in wastewater. Part of this fossil carbon is transformed into sludge. Neglect of direct $CO_2$ emissions causes an under estimation of the carbon footprint for SS treatment and disposal [39–41]. Therefore, direct $CO_2$ emissions were still not accounted for in the carbon footprint but listed in the inventory.

### 2.2.3. Uncertainty Analysis

The data used in this analysis were from existing lab studies and projects. For comparison, some assumptions were made. Uncertainties always exist due to the variability and availability of these data. In general, the original research did not evaluate the uncertainties of the data and results. To further test the accuracy of this study, the data was assumed to follow a normal distribution, and the accounting results were modeled for 100 iterations using a Monte Carlo simulation [42].

### 2.3. Energy Consumption

Thermochemical conversion processes are usually energy-intensive processes because they need to heat raw materials to a high temperature. In order to describe the relationship between energy products (fuel gas, bio-oil, biochar and et al.) and energy consumption, an indicator called the energyconsumption ratio (ECR) was brought in [43–45]. The definition of ECR is as follows:

$$ECR = E_I / E_P, \tag{1}$$

where $E_I$ is the energy input of thermochemical conversion processes (MJ/t) and $E_P$ is the energy production during the processes (MJ/t).

$E_P$ is estimated by the higher heating value (HHV) of energy products and their yields (Y). Taking into account the rate of energy loss and heat recovery, the energy required for the thermochemical conversion processes is:

$$E_P = C_P \times m \times dT \times (1 - R_h) / R_c, \tag{2}$$

where m is the mass of the SS (t), $C_p$ is the specific heat capacity of the SS (kJ/t K), dT is the temperature increase before and after conversion (K, assuming the initial temperature is 298 K), $R_h$ is the heat recovery rate (assuming to be 0.5), and $R_c$ is the combustion efficiency (assumed to be 0.7).

According to [31,46] and by applying Kopp'srule, ECR was calculated using the following equations.

For the HTL conversion process:

$$ECR_{HTL} = [W_i C_{pw} + (1 - W_i) C_{ps}] \times dT \times m \times (1 - R_h) / Y \times HHV \times R_c, \tag{3}$$

where $W_i$ (%) is the moisture content of SS before the HTL process, Y(t) and HHV (kJ/t) are the energy products' yield and higher heating value, respectively, $C_{pw}$ is the specific heat capacity of water (4.18 kJ/kg/K), and $C_{ps}$ is the specific heat capacity of sludge, which can be calculated by:

$$C_{ps} = 0.709\ \alpha + 14.304/2\ \beta + 1.04/2\ \gamma + 0.71\ \delta + 0.918/2\ \varepsilon, \tag{4}$$

where $\alpha$, $\beta$, $\gamma$, $\delta$, $\varepsilon$ represent the mass fractions of C, H, N, S, and O of the SS respectively, and 0.709, 14.304, 1.04, 0.71, 0.918 are the heat capacities of those elements at 298 K (kJ/kgK).

For the pyrolysis conversion process:

$$ECR_{pyro} = [W_v \times C_{pw} \times 75 + W_v\ L_{vap} + (1 - W_v)\ C_{ps} \times dT] \times m\ (1 - R_h)/Y \times HHV \times R_c, \tag{5}$$

where $L_{vap}$ is the latent heat of volatilization for water in SS (2260 kJ/kg) and $W_v$ (%) is the moisture content that is needed to evaporate prior to conversion.

For incineration conversion in this study:

$$ECR_{Inci} = \sum m_i\ Q_i\ /\ Y \times HHV, \tag{6}$$

where $m_i$ is the mass of auxiliary fuels (kg) and Qi is the heat value of auxiliary fuels (kJ/kg).

ECR < 1 indicates that more energy is recovered by-products than energy input in the process; otherwise, more energy is required than is recovered.

### 2.4. Element Balance Analysis

The carbon and nitrogen element balance analysis was carried out based on the elemental composition of sludge and products. C, H, N, S, and O in SS, bio-oil, and biochar samples were analyzed using an elemental analyzer. Fuel gas composition was determined using gas chromatography. Water phase samples were analyzed for total organic carbon and total nitrogen content based on standard methods [7,18,28].

## 3. Results

### 3.1. Carbon Footprint

### 3.1.1. Life Cycle Inventory

The technical parameters of energy consumption, material consumption, products yield, and pollutant emissions were obtained from the literature. For ease of calculation and comparison, some assumptions were made in the life cycle inventory analysis. All the input and output data are displayed in Table 1. Please note that although the direct $CO_2$ emissions of Cases 1 and 3 are listed, the carbon footprint in the next section does not include them. This is because this research followed IPCC guidelines, which consider direct $CO_2$ emission from SS disposal as biogenic and not counted towards GHG emissions.

### 3.1.2. Carbon Footprint

As Figure 2 displays, the HTL, pyrolysis, and incineration processes for SS have a total carbon footprint of 172.50, 322.23, and 242.02 kg $CO_{2eq}$/t SS, respectively. Compared with regular sludge disposal technologies (sanitary landfill of 317.518 kg $CO_{2eq}$/t SS, building materials application of 247.922 kg $CO_{2eq}$/t SS, anaerobic digestion with land application of 190.038 kg $CO_{2eq}$/t SS, and aerobic composting with land application of 146.276 kg $CO_{2eq}$/t SS), thermochemical conversion processes emit similar or smaller amounts of GHG [50]. It is worth noting that these results for traditional SS disposal routes were calculated on a dry sludge basis; the GHG emission associated with SS/biosolid dewatering and drying was not included. As we all know, the drying process takes a huge amount of energy, about 1.808 GJ/t SS (80 wt% of moisture content), equivalent to a GHG emission of 164.17–473.04 kg $CO_{2eq}$ [51,52]. If counted as part of emissions, the thermochemical conversion technologies for SS show great potential for carbon reduction.

**Table 1.** Life cycle inventories of SS disposal processes.

| Item | Unit | Case 1 (HTL) [7,47–49] | Case 2 (Pyrolysis) [27,28] | Case 3 (Incineration) [18,27] |
|---|---|---|---|---|
| Energy input | | | | |
| Electricity | kWh/t SS | 251.8729 | 393.5651 | No electricity added |
| Heat | MJ/t SS | No heat needed | No heat needed | 462 |
| Auxiliary fuel | | | | |
| Coal | ton/t SS | No auxiliary fuel added | No auxiliary fuel added | 0.513 |
| Diesel | kg/t SS | | | 0.055 |
| Chemicals input | | | | |
| $H_2O$ | ton/t SS | No chemical added | No chemical added | 5 |
| CaO | kg/t SS | | | 6 |
| Direct gas emissions | | | | |
| $CO_2$ | kg/t SS | 31.8492 | No gas directly emitted | 54.5 |
| $CH_4$ | kg/t SS | 0.162 | | No $CH_4$ directly emitted |
| Energy output | | | | |
| Electricity | kWh/t SS | No electricity produced | No electricity produced | 53 |
| Fuel gas | MJ/t SS | No fuel gas produced | 1086.8 | No fuel gas produced |
| Biochar | MJ/t SS | Biochar was landfilled without energy recovery | 488 | Biochar was landfilled without energy recovery |
| Bio-oil | MJ/t SS | 2669.16 | 1250.64 | No bio-oil produced |

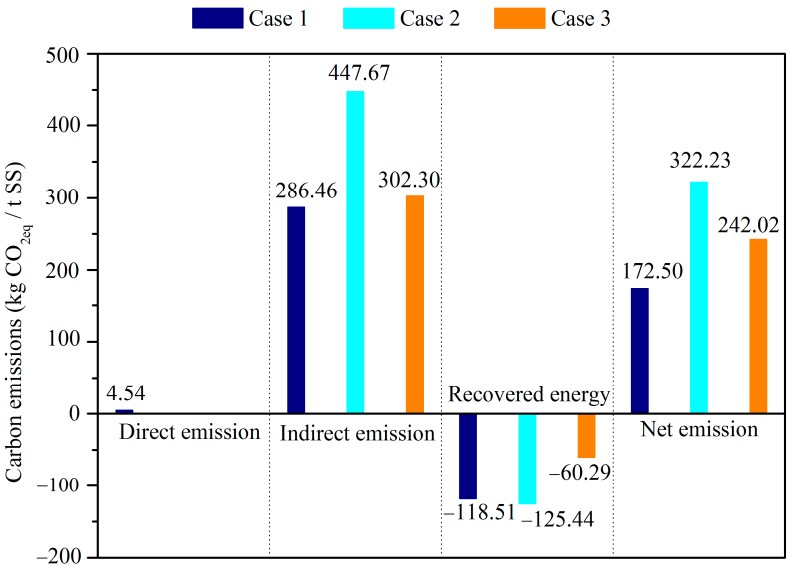

**Figure 2.** Carbon footprints of the three cases.

As the reason described in Section 3.1.1., the SS incineration process actually emitted 242.02 plus 54.5 kg $CO_{2eq}$/t SS, which was very close to the carbon footprint of the SS pyrolysis process. This is consistent with the fact that auxiliary fuel was converted into $CO_2$ through the incineration process. As for the reason why pyrolysis and incineration have similar carbon footprints and larger footprints than the HTL process, it is due to the fact that SS drying takes a huge amount of energy and HTL is more energetically favorable for handling feedstock with a high moisture content [30].

Direct emission reflects the carbon emission effect in situ. Direct emission of the three cases amounted to 4.54, 0, and 0 kg $CO_{2eq}$ when 1 t dewatered sludge was disposed of. Case 2 showed the best carbon reduction potential compared to cases 1 and 3 with no GHG emission directly into the environment. This is because the carbon in SS had been recovered in the form of energy or fixed in biochar through pyrolysis conversion, instead of in the

form of fugitive emissions or partly fugitive emissions into the atmosphere in case 2 and case 3.

However, for the three cases, more of the carbon footprints were derived from indirect emissions as Figure 2 demonstrates. Indirect emissions amounted to 286.46, 447.67, and 302.30 kg $CO_{2eq}$, respectively, which were related to energy and chemical consumption during the conversion process. It is easy to understand that thermochemical conversion processes are energy-intensive processes and need much energy to break sludge particles into small molecules. Case 2 demanded more energy than case 1 because the conversion process needed to pre-dry the feedstock to remove water content and heat sludge to a higher temperature to purchase valuable products. Indeed, the three cases recovered energy during the thermochemical conversion processes when sludge was disposed of, which could offset the total emissions of 118.51, 125.44, and 60.29 kg $CO_{2eq}$, respectively. However, the recovered energy was not enough to offset the energy input and the related GHG emissions. For the local place, the carbon footprint of sludge disposal could be cut down through the pyrolysis process. From a fulllife cycle and global perspective, the carbon footprint had not been reduced; it was just moving from sludge disposal plants to power stations, chemical plants and et al. It does not mean that the thermochemical conversion process is of no benefit at all. Concentrated discharge of GHG in one place is easier to manage and treat than disorganized discharge in many places. Some measures can be taken to minimize the GHG emission of power stations, chemical plants and et al., such as energy efficiency improvements and carbon capture. Accordingly, the GHG emission of the downstream process can be reduced.

Meanwhile, a sensitivity analysis was carried out by modifying the relevant input and output from 90% to 110% to check its impact on the carbon footprint. Input energy, direct emission, and recovered energy were chosen as the variables. Figure 3 presents the sensitivity analysis result of the three cases.The higher the slope of the line, the higher the sensitivity.

From Figure 3, it can be seen that for the three cases, the carbon footprint was most sensitive to input energy, and the recovered energy took second place. The carbon footprint was least sensitive to direct emissions. This result was consistent with the previous discussion.

### 3.1.3. Carbon Footprint under Green Electricity

As illustrated above, GHG emissions derived from energy consumption are the major contributor to the total carbon footprint for SS thermochemical conversion processes. Most energy consumption during the thermochemical conversion process is in the form of electricity. In order to figure out how the energy source affects its GHG emission characteristic, some scenarios were assumed and the carbon footprints were analyzed. Four scenarios of the electricity sector carbon reduction rate, i.e., 20%, 50%, 70%, and 100%, were examined. The results are demonstrated in Figure 4.

A decreasing trend of indirect emissions, as well as net emissions with carbon reduction by the electricity sector, was observed in Cases 1 and 2. When the electricity sector cut down carbon emissions by 70%, Cases 1 and 2 almost reached carbon neutrality. Case 3 had a constant indirect emission and carbon footprint due to the auxiliaryfossil energy utilization in the incineration process, such as coal and diesel. This part of the GHG emissions could not be offset by carbon reduction in the electricity sector. Therefore, the carbon emission characteristics of sludge thermochemical conversion may be strongly affected by its energy resources. With the rapid development of clean energy, thermochemical conversion technology will release great potential in carbon reduction for SS management.

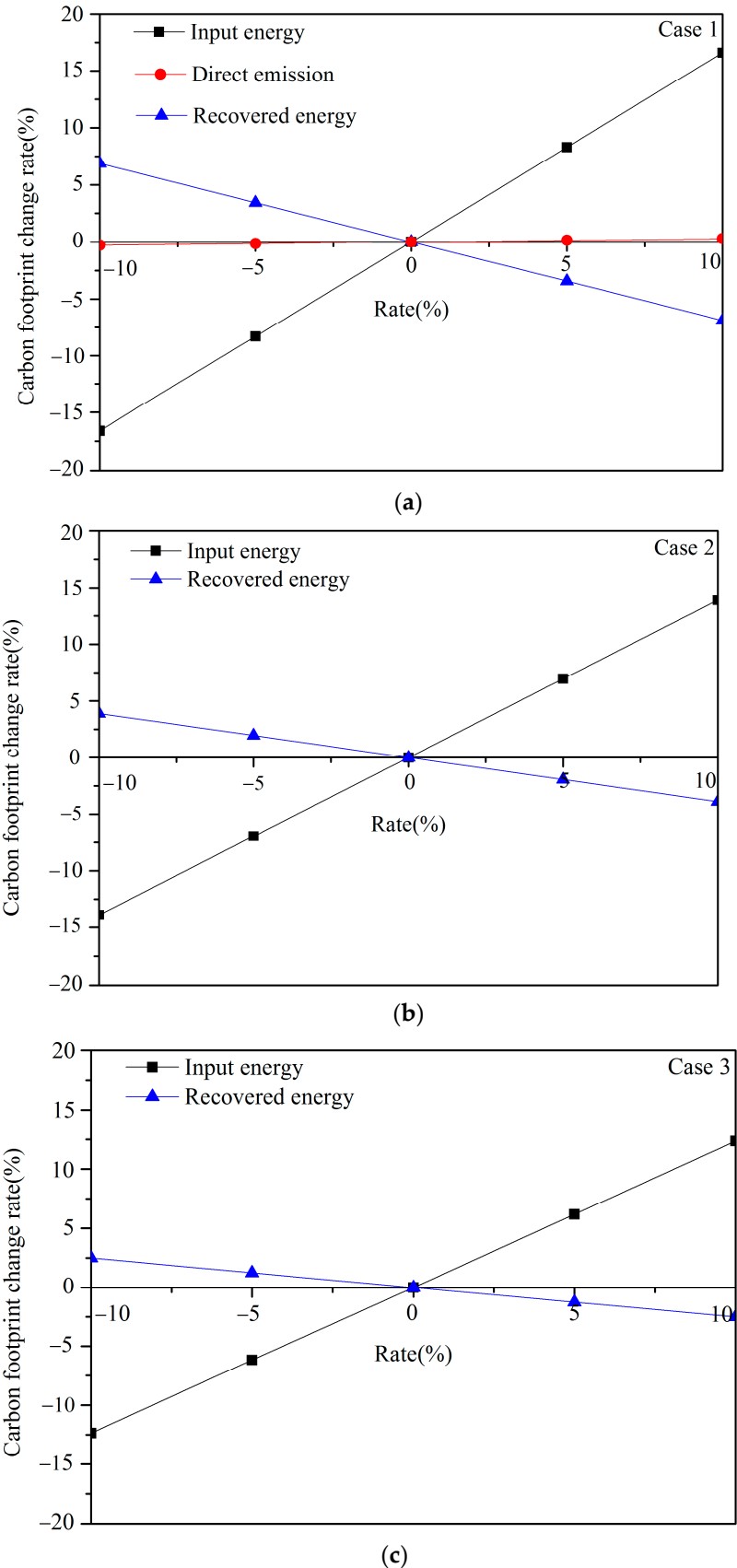

**Figure 3.** Sensitivity assessment for Case 1 (**a**), Case 2 (**b**), and Case 3 (**c**).

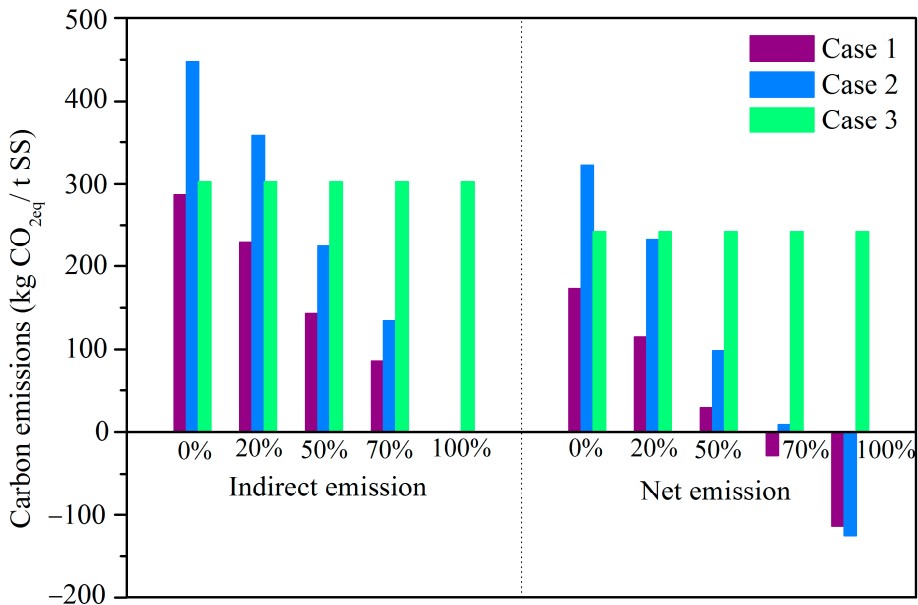

**Figure 4.** Carbon footprint of the three cases with different electricity carbon reduction rates.

3.1.4. Uncertainty Analysis

Uncertainty analysis was conducted by modifying the relevant inputs or outputs of the analyzed cases. For illustration, Figure 5 presents the uncertainty analysis result of the carbon footprint based on changes in energy inputs, direct emissions, and energy outputs. The uncertainties of the carbon footprints are exhibited as standard deviation bars.

It can be seen that the uncertainty of input energy in Case 3 was the largest. Thismay be due to the three kinds of input energy. Their impacts on the results may be added and amplified, while the other two cases had only one kind of input energy. Uncertainties of the other cases were below 30%, which indicates that the results are highly credible.

*3.2. Energy Consumption*

Based on the definition and calculation method, an ECR ratio > 1 indicates that the whole process is a net energy consumer, while an ECR ratio < 1 means that the process produces more energy than it consumes. In previous research, ECR was applied in order to study the efficiency of the thermochemical process for wet biomass, such as algal feedstock. The HTL and pyrolysis processes resulted in 0.44–4 and 0.92–1.24 of ECR based on the different kinds of algae and the experimental conditions. It can be seenfrom the results in Table 2 that Case 1 and Case 2 had positive energy consumption ratios when this calculation method was used, which indicated that the systems were net energy producers. The HTL and pyrolysis processes showed more favorable energy conversion efficiency when SS was used as feedstock. From the view of energy conversion, HTL and pyrolysis process have feasibility for SS disposal. Case 1 had a relatively higher efficiency because it did not need to pre-dry the feedstock, which reduced the energy needed to evaporate the extra water content. The heating temperature was also relatively low compared with the pyrolysis process.

**Table 2.** ECR ratio of the three cases.

|  | **Case 1-HTL** | **Case 2-Pyrolysis** | **Case 3-Incineration** |
|---|---|---|---|
| EI(kJ) | 906,742 | 1,416,834 | 11,190,150 |
| EP(kJ) | 2,669,160 | 2,825,440 | 190,800 |
| ECR | 0.34 | 0.50 | 58.65 |

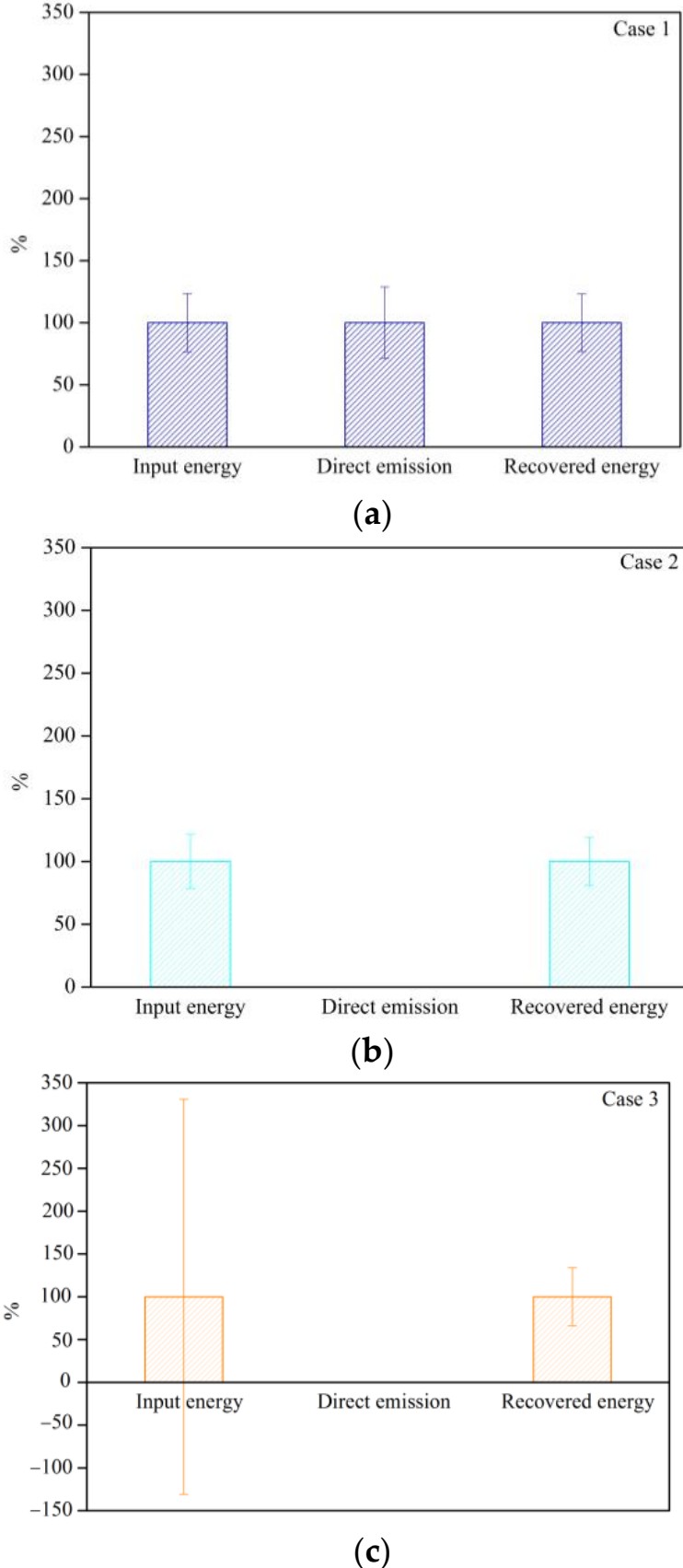

**Figure 5.** Uncertainty analysis of Case 1 (**a**), Case 2 (**b**), and Case 3 (**c**) (by Monte Carlo simulation).

Since feedstock moisture is crucial to the efficiency of the thermochemical process, the impacts of moisture on ECR were studied. According to previous research, the ECR ratio increased with moisture content in the HTL reactor. This is because water has a high specific heat capacity and the energy demand grows sharply when moisture content increases. However, the product yield is not impacted by a feedstock loading increase with the same proportion of water [53]. Therefore, ECR decreases if the feedstock loading increases, due to an increase in energy products obtained. The impact of moisture on ECR is shown in Figure 6, which indicates that an increase in moisture results in an increase in ECR and hence a decrease in energy conversion efficiency.

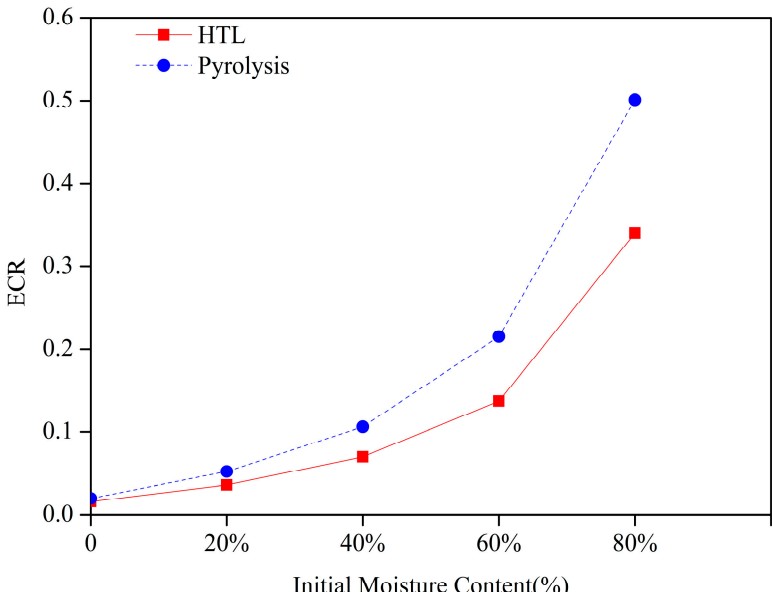

**Figure 6.** ECR for HTL and pyrolysis of SS with varying initial moisture contents.

The ECR ratio of Case 3 indicates that the system consumed more energy than it produced. This is probably attributed to the high water concentration and low heating value of SS, which makes it need a large quantity of auxiliary fuel to be combusted.

### 3.3. Element Balance Analysis

The transport of carbon and nitrogen and their distribution in output streams is important, as this influences not only the quality of the products but also the GHG emission performance. For instance, it is desirable for a process to concentrate most of the carbon and nitrogen in the products that can be reused instead of wasting them during processing.

Figure 7 presents the carbon and nitrogen conversion ratio during the HTL, pyrolysis, and incineration processes of SS. For case 1, bio-oil recovered around 60 wt% of carbon and 36 wt% of nitrogen in the SS. 10 wt% of carbon and 8.5 wt% of nitrogen migrated to the solid phase. The great majority of nitrogen content in the initial feedstock (around 58 wt%) was found in the water phase, while its carbon content was lower (about 20 wt%). The rest of the carbon, about 10 wt%, was converted into $CO_2$ and $CH_4$ in the gas stream, while no nitrogen was found in the gas phase.

Case 2 was shown to recover 54 wt% of carbon through bio-oil, 33.78 wt% by fuel gas, and 12.22 wt% through biochar. For nitrogen, the majority was transferred into bio-oil at 77.65 wt%, followed by bio-char at 22.35 wt%. Once again, no nitrogen was found in the gas phase.

For case 3, most of the carbon and all of the nitrogen migrated to flue gas. Only a very small part of the carbon is found in the fly ash and slag [54]. Through the sludge incineration process, the carbon in the sludge was converted to CO, $CO_2$, PCDD (polychlorinated dibenzopdioxin), and PAHs (Polycyclic Aromatic Hydrocarbons) et al., while the nitrogen

was converted to NOx and $N_2O$. This means that SS incineration may cause secondary pollution and that the flue gas needs to be processed carefully.

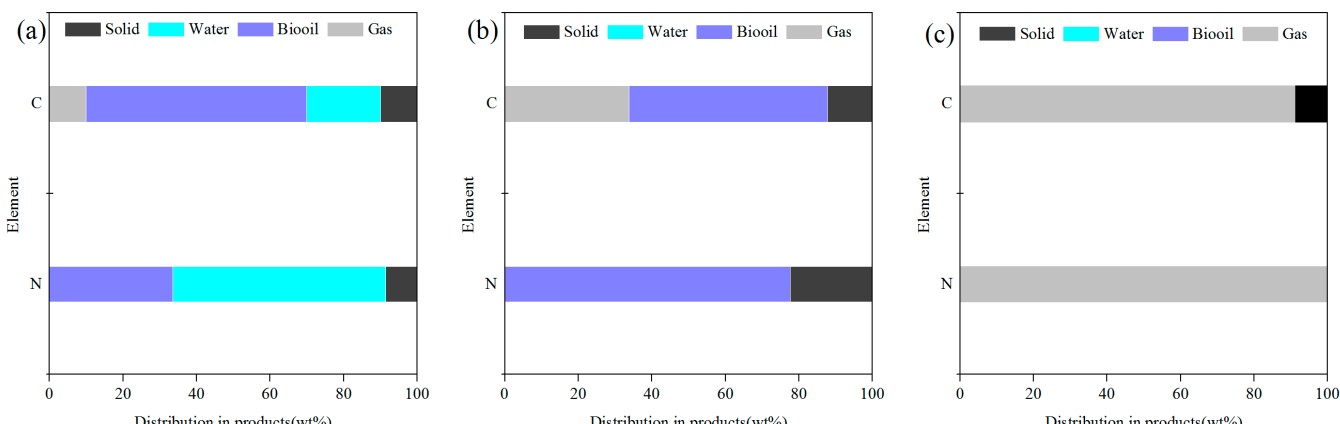

**Figure 7.** Distribution of carbon and nitrogen in the output streams for Case 1 (**a**),Case 2 (**b**), and Case 3 (**c**).

In the sludge anaerobic digestion process, only 50 wt% of volatile solid can be converted to biogas, and biogas contains 65 wt% $CH_4$ and 35 wt% $CO_2$. The carbon recovery rate is no more than 25 wt% [55–57]. Compared with anaerobic digestion, 60 wt% carbon efficiency to bio-oil in the HTL process and 100 wt% carbon to valuable products in the pyrolysis process is very appealing.

The wastewater of the HTL process needed to be further processed since there was still a large quantity of carbon and nitrogen to be recovered. Integration of anaerobic digestion and HTL is a promising and economic approach. It was reported that around 60 wt% of organics in HTL waters from algae and modeled food waste was removed through anaerobic digestion [58,59]. The molecular composition of HTL wastewater from SS and its transformation during anaerobic digestion were studied and will contribute to the development of efficient methods for HTL wastewater treatment in the future [60].

Meanwhile, the integration of HTL and microalgae could be a promising avenue. There is some research that explores carbon and nutrient capture by microalgae to valorize wastewater of biomass HTL process [61]. However, little research on the valorization of refractory wastewater from the SS HTL process was reported, maybe due to its complex components and inhibiting effect on biomass growth. Additionally, the gas phase of HTL is mainly composed of $CO_2$, which should be collected and recycled when in large-scale application. Therefore, valorizing the wastewater and gas of SS HTL to recover carbon and other nutrients is an important step, not only clearly beneficial for carbon reduction, but also good for energy recovery.

It should be noted that this analysis is based on dry sludge, and does not account for carbon and nitrogen loss with leachate from the sludge drying process, which may influence the results.

## 4. Conclusions

This paper analyzed the carbon footprints, energy consumption, and element balances of SS thermochemical conversion technologies. HTL, pyrolysis, and incineration processes of SS result in carbon footprints of 172.50, 322.23, and 242.02 kg $CO_{2eq}$/t SS, respectively. The input energy is identified as a major GHG contributor, and significant differences were observed in the energy consumption (906.742, 1416.834, and 11190.150 MJ/t SS, respectively). HTL and pyrolysis technology of SS are environmentally friendly and a strategic alternative to mitigate GHG emissions, especially in the scenario of green electricity as an input energy. HTL shows the best carbon reduction potential with the least carbon footprint compared to traditional SS disposal routes and the other two thermochemical

conversion technologies. HTL also has the most favorable ECR ratio, which means that it is feasible from the view of energy conversion efficiency. For future research, the need for an effective approach to recover carbon in the water and gas phase of the HTL process is urgent.

This research demonstrates clearly the potential for thermochemical conversion technologies to fight against global warming. It provides a useful reference for technology options and policy-making in the SS management sector through quantitative information. It could serve as a material and methodological reference for other studies dealing with SS management and carbon footprint analysis.

**Author Contributions:** Conceptualization, L.L. and B.Y.; methodology, G.D.; software, L.L.; validation, L.L.; formal analysis, L.L.; investigation, Y.D.; resources, H.L. and J.S.; data curation, Y.W.; writing—original draft preparation, L.L.; writing—review and editing, Y.S. and T.H.P.; visualization, W.L.; supervision, G.C.; project administration, Y.Z.; funding acquisition, Y.Z. All authors have read and agreed to the published version of the manuscript.

**Funding:** This research was funded by National Key R & D Program, China (No. 2018YFE0106400).

**Institutional Review Board Statement:** Not applicable.

**Informed Consent Statement:** Not applicable.

**Data Availability Statement:** Not applicable.

**Conflicts of Interest:** The authors declare no conflict of interest.

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
