# Peer review of "Carbon Footprint Analysis of Sewage Sludge Thermochemical Conversion Technologies"

_sustainability, doi:10.3390/su15054170_

Round 1

Reviewer 1 Report

Minor Revision

1-abstract

The methodology is not defined. (Open LCA)

2-Introduction

literature related to LCA not enough.

2-Material and Method

Functional unit

System boundary (Gate to Gate)

3-Results

-          Figure 7 will be more cleared if it is in one figure.

-          Global warming can be calculated using other software (SimaPro,…)

Reviewer 2 Report

I personally consider the topic relevant in that area. It's not a new topic, but this paper on that topic, in my opinion, is for publication.

I believe that the methodology and the entire work is correctly written. It doesn't need any improvements.
The results, discussion and conclusions are in accordance with the set goal of the work.

The references used are appropriate to the content. Tables and figures are consistent with the text and discussion. 

Summary is too long. Theory does not belong in the summary.

Reviewer 3 Report

The motivation for this study is, presumably, to provide policymakers with a side-by-side comparison of three, plausible approaches to upvaluing sewage sludge: hydrothermal liquefaction and pyrolysis to make precursors to liquid fuels, and incineration to generate heat, which would likely be used to make steam for district heating and or generation of electricity. 

Many of the details required to evaluate or replicate the lifecycle analyses are, however, missing. The assumptions about the disposal of the wastewater from HTL should be described more fully—the reactor cannot be operated closed loop as depicted in Figure 1 because there will need to be some blow down. Will the excess be returned to the WWTP? Will it be injected in a storage well? What is the fate of the gas made during HTL and pyrolysis? The heating value of the bio-oil produced by HTL cited in Table 1 seems to me to be a bit low. More typically, HTL captures a very large fraction of the heating value of the input solids [see, for example, Snowden-Swan, L. J.; Zhu, Y.; Jones, S. B.; Elliott, D. C.; Schmidt, A. J.; Hallen, R. T.; Billing, J., M.; Hart, T., R.; Fox, S. P.; D., M. G. Hydrothermal Liquefaction and Upgrading of Municipal Wastewater Treatment Plant Sludge: A Preliminary Techno-Economic Analysis, Pacific Northwest National Laboratory, 2016].  Why wouldn’t the solids from HTL and Pyrolysis be burned for process heat or sold as fuel? What is the amount and type of auxiliary fuel assumed for the incineration (coal? Natural gas?) It would be worth commenting on the effect of the wetness of the feed—it is not at all clear or known to me that “SS with moisture content of 40 wt% shows almost the same combustion characteristic as the 10 wt% ones”. 

It would be good to explain why, given that incineration converts the SS plus the auxiliary fuel into carbon dioxide, how it can exhibit an intermediate set of emissions.

Is there an opportunity to use the off gas from the WWTP as a fuel to heat either the HTL or the pyrolysis reactor?

A minor comment, “prepositive” is not used correctly. I think that the authors were confused by their dictionary. From context, it seems that the authors meant to use the word “prior”.

Also it seems that authors neglected to learn from Lozano, et al., https://doi.org/10.1016/j.ecmx.2022.100178.
